# FOCUS ON PRIMARY: DIFFERENTIAL DIVERSE DATA AUGMENTATION FOR GENERALIZATION IN VISUAL REINFORCEMENT LEARNING

## ABSTRACT

In reinforcement learning, it is common for the agent to overfit the training environment, making generalization to unseen environments extremely challenging. Visual reinforcement learning that relies on observed images as input is particularly constrained by generalization and sample efficiency. To address these challenges, various data augmentation methods are consistently attempted to improve the generalization capability and reduce the training cost. However, the naive use of data augmentation can often lead to breakdowns in learning. In this paper, we propose two novel approaches: Diverse Data Augmentation (DDA) and Differential Diverse Data Augmentation (D3A). Leveraging a pre-trained encoder-decoder model, we segment primary pixels to avoid inappropriate data augmentation affecting critical information. DDA improves the generalization capability of the agent in complex environments through consistency of encoding. D3A uses proper data augmentation for primary pixels to further improve generalization while satisfying semantic-invariant state transformation. We extensively evaluate our methods on a series of generalization tasks of DeepMind Control Suite. The results demonstrate that our methods significantly improve the generalization performance of the agent in unseen environments, and enable the selection of more diverse data augmentations to improve the sample efficiency of off-policy algorithms.

## 1 INTRODUCTION

Reinforcement Learning (RL) has achieved remarkable success in many fields, such as robot control (Kalashnikov et al., 2018), financial trading (Hambly et al., 2023), and autonomous driving (Kiran et al., 2021). Image-based RL changes the form of obtaining observations from states to images, which is more in line with the real world. Therefore visual RL has been widely used in areas such as video games (Mnih et al., 2015) and autonomous navigation (Zhu et al., 2017). However, a critical challenge is how an agent overfitted in a single training environment can still achieve good performance when encountering unseen environments (Kirk et al., 2023).

In RL, the agent continuously collects samples from a single training environment for training to increase the reward obtained. This process has a tendency to overfit the current training environment, resulting in performance degradation or even failure when tested or deployed in other environments. Data augmentation plays a crucial role in the success of computer vision, and also has wide applications in supervised learning (Cubuk et al., 2019; 2020), semi-supervised learning (Berthelot et al., 2019) and self-supervised learning (Chen et al., 2020). By using multiple augmented views of the same data as input, encoder works on learning consistency in their internal representation, with the goal of learning a visual representation that improves generalization and data efficiency (Henaff, 2020; Dunion et al., 2023).

Diverse data augmentation is expected to be used to improve the sample efficiency of visual RL. This is because pixel-level transformation of images can generate more samples, expanding the sample space. Since Laskin et al. (2020b) firstly use data augmentation to improve the generalization of visual RL, many works have attempted to apply diverse data augmentation to visual RL (Kostrikov et al., 2021; Hansen et al., 2021b). In fact, not all data augmentation methods benefit the agent in the

same way, and even many strong data augmentations cause the semantics of the observation images to change, which can lead to the failure of training.

The capability of humans to learn and adapt quickly to changes in vision is partly due to being more focused on the primary part. For example, humans can easily recognize the object is a bird if it is obvious in the background. But, if the object is occluded or almost the same color as the background, it would be difficult for humans to recognize, as shown in Figure 1. In visual RL, the high-dimensional par-

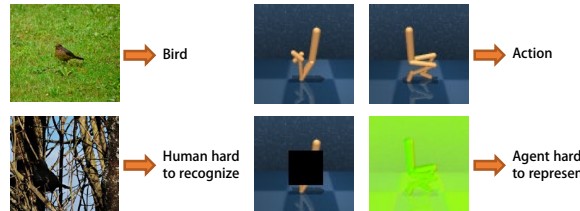

Figure 1: Humans have difficulty recognizing objects whose colors are very close to the background. Agents are similarly unable to represent naive strong augmentation well.

tially observed image is first fed into an encoder, which encodes the high-dimensional image into low-dimensional embedding, similar to the way humans acquire visual information through their eyes. Therefore, when improving the generalization of an observed image by different data augmentation, it should not be naive to use data augmentation to transform each pixel, which can lead to unstable training and bad generalization (Yuan et al., 2022a).

We propose an encoder-decoder structure based on Segnet (Badrinarayanan et al., 2017). This model segments the observation primary from the background in an image, following the behavioral intuition of human learning and generalization. Different treatments for primary and background, such as different kinds and hardness data augmentation, can be realized. Specifically, we use no or slight pixel-level data augmentation for the primary region as well as diverse, aggressive data augmentation for the irrelevant background region. This distinctive use of data augmentation can help the agent focus on the primary information, with better assurance of semantic-invariance of transformation, and thus improve the generalization capability. We propose two framework variants, **D**iverse **D**ata **A**ugmentation (DDA) and **D**ifferential **D**iverse **D**ata **A**ugmentation (D3A).

Our main contributions are summarized as follows:

- We use a simple clustering algorithm for image segmentation based on the color and location information of the image and construct the DMC Image Set. With this dataset, we pre-train an encoder-decoder model for segmenting out the visual primary regions of the observation images.

- We propose diverse data augmentation that preserves the primary pixel and applies multiple random data augmentations to irrelevant background pixels, and it can improve the generalization performance of the agent.

- We propose differential diverse data augmentation, which quantitatively measures the distance in Q-values of augmented observations and retains acceptable augmented observations. While applying random augmentations to the irrelevant background and slight augmentations to the primary pixels under satisfying semantic-invariant state transformation.

- We conduct extensive experiments in DMControl Generalization Benchmark (DMC-GB) (Hansen & Wang, 2021). The agent is trained in a single environment and generalization performance is evaluated in three environments: color-hard, video-easy, and video-hard. Compared to the state-of-the-art methods, our methods achieve more advanced performance and sample efficiency when encountering complex and unseen environments.

## 2 RELATED WORKS

**Representation Learning in RL.** Agent in RL can show great potential in complex tasks. However, current RL algorithms generally require frequent interactions with the environment to generate a large amount of data. The data acquired by visual RL are high-dimensional images that need to be encoded into low-dimensional vectors by encoder. Therefore the merit of representation learning directly affects the performance. One of the more widely researched directions is to combine self-supervised learning with RL (Laskin et al., 2020a). Some more recent works attempt to introduce the pre-trained model into RL to improve representation learning (Yuan et al., 2022b). Zhang et al.

(2022) develop a policy pre-training method by learning from driving videos on the web, and then transfer the pre-trained representations to the reinforcement learning task of visual driving.

**Generalization in Visual RL.** Previous works focus on improving the generalization performance of visual RL in the following three ways. (1) Improve the similarity between training and testing environments, such as domain randomization and data augmentation. Domain randomization aims to maximally cover the distribution of the test environment during training (Chebotar et al., 2019). In fact, it is a sample inefficient training method, and it is impossible to generate all the possible environments. Data augmentation transforms and expands the training data during the training process, which can simulate the changes and noise in the real scene to improve the generalization capability of the model. (2) Add auxiliary tasks. Hansen & Wang (2021) learn generalization by maximizing the similarity between latent representations of augmented and non-augmented images. Dunion et al. (2023) introduce an auxiliary task of self-supervised learning to disentangle image representations. (3) Regularization explicitly imposes constraints or penalties on the parameters of the model to prevent overfitting (Cobbe et al., 2019).

**Data Augmentation for RL.** Data augmentation can expand the sample space and produce more equivalent data without substantially increasing the sample sampling time. It can effectively make up for the lack of training data, prevent the model from overfitting. Laskin et al. (2020b) conduct the research on RL using data augmentation on pixel-based and state-based inputs. Fan et al. (2021) train with weak augmentation to obtain the expert policy, and then distill the student policy that could accept strong augmentation. Hansen et al. (2021b) further point out the cause of instability caused by the use of data augmentation in off-policy RL, and propose only augmenting the observation of the current time step to reduce the variance of Q-value targets. Inappropriate data augmentation can change the semantics of observations and lead to training divergence. Yuan et al. (2022a) determine the association between each pixel point and the action by calculating the Lipschitz constant for that pixel, retaining task-relevant pixels from being augmented. This is also similar to our starting point. However, they do not consider additional data augmentation options, and the computational complexity and computation time are less acceptable.

## 3 BACKGROUND

**Problem formulation.** We consider problem in a Markov Decision Process (MDP) (Bellman, 1957) formulated by the tuple $\langle \mathcal{S}, \mathcal{A}, r, \mathcal{P}, \gamma \rangle$. $\mathcal{S}$ is the state space. $\mathcal{A}$ is the action space. $r : \mathcal{S} \times \mathcal{A} \to \mathbb{R}$ is the reward function, $\mathcal{P}(s_{t+1} \mid s_t, a_t)$ is the state transition function, $\gamma \in [0, 1)$ is the discount factor. The goal of RL is to learn an optimal policy $\pi^* = \text{argmax}_\pi \mathbb{E}_{a_t \sim \pi(\cdot \mid s_t), s_t \sim \mathcal{P}} \left[ \sum_{t=0}^{T} \gamma^t r(s_t, a_t) \right]$, that maximizes expected sum of discounted future rewards. In visual RL, the agent obtains image-based observations from the environment, so the tuple is described as $\langle \mathcal{O}, \mathcal{A}, r, \mathcal{P}, \gamma \rangle$, where $\mathcal{O}$ is the high-dimensional observation space. Crucially, we expect to generalize to unseen environments settings $\langle \mathcal{O}', \mathcal{A}, r, \mathcal{P}, \gamma \rangle$, where $\mathcal{O}'$ is different from $\mathcal{O}$.

**Deep Q-learning.** Model-free off-policy RL aims to estimate an optimal state-action value function $Q^* : \mathcal{S} \times \mathcal{A} \to \mathbb{R}$ as $Q_\theta(s, a) \approx Q^*(s, a) = \max_{\pi_\theta} \mathbb{E}[R_t \mid s_t = s, a_t = a]$ using function approximation. In practice, this is achieved by means of the single-step TD-error $(r(s_t, a_t) + \gamma \max_{a_{t+1}} Q_{\tilde{\theta}}(s_{t+1}, a_{t+1})) - Q_\theta(s_t, a_t)$ (Sutton, 1988), where $\tilde{\theta}$ parameterizes a target state-action value function. We choose to minimize this TD-error directly using a mean squared error loss, which gives us the objective,

$$\mathcal{L}_Q(\theta, \psi) = \mathbb{E}_{s_t, a_t, s_{t+1} \sim \mathcal{B}} \left[ \frac{1}{2} \left[ \left( r(s_t, a_t) + \gamma \max_{a_{t+1}} Q_{\tilde{\theta}}(s_{t+1}, a_{t+1}) \right) - Q_\theta(s_t, a_t) \right]^2 \right], \quad (1)$$

where $\mathcal{B}$ is a replay buffer with transitions collected by the policy. We can derive a greedy policy directly by selection actions $a_t = \arg\max_{a_t} Q_\theta(s_t, a_t)$. We provide details of other base algorithms of RL used, such as SAC and SVEA in Appendix A.

**Definition 1** *(Optimality-Invariant State Transformation* (Kostrikov et al., 2021)*)*

Given a MDP, we define a transformation $f : \mathcal{O} \times \mathcal{T} \to \mathcal{O}'$ as an optimality-invariant transformation if $\forall o \in \mathcal{O}, a \in \mathcal{A}, \nu \in \mathcal{T}$ where $\nu$ are the parameters of $f(\cdot)$, drawn from the set of all possible parameters $\mathcal{T}$ satisfies:

$$Q(o, a) = Q(f(o, \nu), a). \qquad (2)$$

Although data augmentation potentially improves generalization to unseen observation spaces in training, it is likely to be detrimental to Q-value estimation (Stooke et al., 2021; Schwarzer et al., 2020). Therefore, the optimality-invariant state transformation attempts to define that the Q-value of the critic output is the same as the original Q-value after adding perturbations to the original observation or transforming by some augmentation method. We further illustrate optimality-invariant state transformation in two concrete aspects:

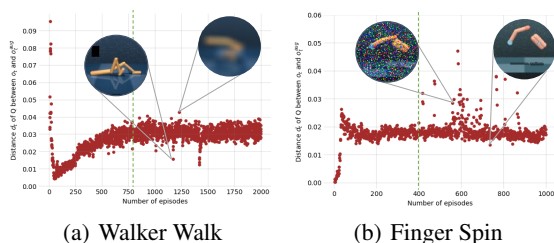

(a) Walker Walk  (b) Finger Spin

Figure 2: Different augmentations result in different distance between the Q-values of the augmented observation and the original observation. Green scale: stabilized training steps, avoiding the impact of early instability on the threshold $\epsilon$

- We expect the added data augmentation to be appropriate, i.e., to satisfy optimality-invariant state transformation. But in fact, this is an idealized definition, since even the smallest perturbations can lead to non-zero Q-value differences. In Figure 2, we show the distance in Q-value estimates between the augmented and original images during training for two tasks in DMC-GB, where the augmentation is randomly selected from 8 different data augmentations. We can find that the distance of Q-value between different augmentation choices is different after training is stabilized, denoted as:

$$d(o, f(o, \nu)) = \frac{|Q(f(o, \nu), a) - Q(o, a)|}{Q(o, a)}, \qquad (3)$$

- Optimality-invariant state transformation is indicated by Q-value estimation that different augmented views of the same observation should be semantically invariant for the agent. We add a threshold $\varepsilon$ to the above optimal-invariant state transformation for the distance in Q-value estimation. The threshold $\varepsilon$ measures the acceptable level of data augmentation, and when $d(o_t, o_t^{aug}) < \varepsilon$, such data augmented views are considered semantically invariant. We define semantic-invariant state transformation, that allows a trade-off between accurate Q-value estimation and generalization when using data augmentation.

**Definition 2** *(Semantic-Invariant State Transformation)*

Given a MDP, we define a transformation $f : \mathcal{O} \times \mathcal{T} \to \mathcal{O}'$ as an semantic-invariant transformation if $\forall o \in \mathcal{O}, a \in \mathcal{A}, \nu \in \mathcal{T}$ where $\nu$ are the parameters of $f(\cdot)$, drawn from the set of all possible parameters $\mathcal{T}$ satisfies:

$$\frac{|Q(f(o, \nu), a) - Q(o, a)|}{Q(o, a)} < \varepsilon. \qquad (4)$$

## 4 METHODS

In order to imitate human visual learning generalization, we propose the Diverse Data Augmentation (DDA) based on a pre-trained semantic segmentation model. It trains the agent to focus on the primary pixels and avoids over-alteration of the observation semantics while introducing diverse data augmentation methods. To further improve the generalization performance for different environments, we propose Differential Diverse Data Augmentation (D3A) , which uses different choices of data augmentation for primary and background. Both DDA and D3A are compatible with any model-free off-policy RL algorithm and directly improve the sample efficiency and generalization performance of visual RL from raw pixels.

### 4.1 ARCHITECTURE OVERVIEW

An overview of the DDA and D3A architecture is provided in Figure 3. First, $o_t$ is fed into the initially pre-trained encoder-decoder model to obtain the 0-1 matrix $M_t$. The observation $o_t$ is

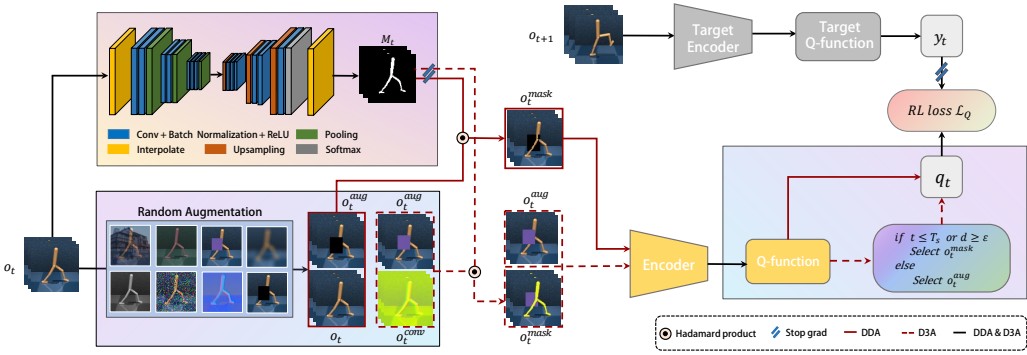

Figure 3: Overview. A batch of transitions $\{o_t, a_t, r_t, d, o_{t+1}\}$ are sampled from replay buffer. $o_t$ has two data streams: one is fed into the segmentation model to get $M_t$, which is then Hadamard producted $\odot$ with different augmented images $o_t^{aug}$ and fed into the encoder of RL. The other is augmented by randomly selecting from a set of data augmentation methods to get $o_t^{aug}$.

also augmented by randomly selecting one from a set of data augmentation methods to obtain $o_t^{aug}$. The solid and dashed red lines in overview represent the different data streams of DDA and D3A, respectively. In DDA, the augmented observation $o_t^{aug}$ and the original observation $o_t$ are Hadamard producted $\odot$ by $M_t$ to obtain $o_t^{mask}$ focusing on the primary. Then $o_t^{mask}$ is fed into the encoder and the Q-network of RL to obtain Q-value, which is then used for the subsequent reinforcement learning process. In D3A, the original observation $o_t$ augmented by default using random convolution to obtain $o_t^{conv}$. $o_t^{conv}$ and $o_t^{aug}$ are Hadamard producted $\odot$ with $M_t$ to get $o_t^{mask}$ of D3A. $o_t, o_t^{aug}$ and $o_t^{mask}$ are fed into the encoder of RL to compute the Q-value and compute the distance $d$ as Equation 3. The Q-value used for the loss of RL is selected by the judgment conditions of D3A. We redefine our objective as

$$\mathcal{L}_Q(\theta) = \mathbb{E}_{o_t, a_t, o_{t+1} \sim \mathcal{B}} \left[ \left\| Q_\theta\left(o_t, a_t\right) - y_t \right\|_2^2 + \left\| Q_\theta\left(o_t^{mask}, a_t\right) - y_t \right\|_2^2 \right], \tag{5}$$

where $y_t = r_t + \gamma \max_{a_{t+1}} Q_{\tilde{\theta}}(o_{t+1}, a_{t+1})$.

The architecture of the encoder-decoder model that obtains the primary pixel positions is detailed in Subsection 4.2, and the resulting mask is also the key and basis for realizing the subsequent diverse and differential data augmentation. The framework for DDA using masks and data augmentation method sets is detailed in Subsection 4.3. The framework for D3A that ensures semantic invariance of observations is detailed in Subsection 4.4.

## 4.2 FOCUS ON PRIMARY PIXELS WITH THE MASK

Based on the deep fully convolutional neural network structure Segnet (Badrinarayanan et al., 2017), we design a smaller encoder-decoder structure for specific environments using 7 encoding and 7 decoding layers, and 1 pixel-level classification layer, which is more lightweight than Segnet. Although the Segnet-basic of the shallow network is slightly smaller than our architecture, our network structure can achieve better results.

Each encoder layer corresponds to a decoder layer, and the encoder consists of convolutional layers, batch normalization layers, ReLU layers, and pooling layers, with the goal of gradually reducing the spatial size of the feature map while extracting higher-level feature information. The convolutional layers generate a set of feature maps by convolution, which does not change the size of the feature maps since the same padding convolution is used. After that, maximum pooling with a $2 \times 2$ non-overlapping window and stride 2 is performed. The maxpooling layer increases the receptive field to reduce the size of the image. At the same time, it records the index position of the maximum value.

The decoder consists of upsampling layers, convolutional layers, batch normalization layers, and ReLU layers. The upsampling layer reduces the input feature map to its original size. The data of the input feature map is restored according to the index position of the encoder pooling layer, and the other positions are zero. The maxpooling layer and the upsampling layer are connected through the max pooling index. The final decoder output is fed into a two-class softmax classification layer. This

---

**Algorithm 1** Diverse Data Augmentation (DDA)

---

**Input:** random initialized network parameters $\theta, \psi$; candidate data augmentation list $\mathcal{F}$; pre-train segmentation model $\phi$; total number of environment steps $T$; batch size $N$; target network update rate $\tau$; learning rate $\lambda$; replay buffer $\mathcal{B}$

**Output:** Loss $\mathcal{L}_Q$ and updated parameters $\theta, \psi$

1: **for** timestep $t = 1, 2, \cdots, T$ **do**
2:      $a_t \sim \pi_\theta(\cdot | o_t)$
3:      $o_{t+1} \sim \mathcal{P}(\cdot | o_t, a_t)$
4:      $\mathcal{B} \leftarrow \mathcal{B} \cup (s_t, a_t, r_t, o_{t+1})$
5:      Sample a batch of size $N$ from $\mathcal{B}$
6:      Random select augmentation $f$ from $\mathcal{F}$
7:      **for** transition $i = 1, 2, \cdots, N$ **do**
8:          Get the primary position mask $M_i = \phi(o_i)$
9:          $o_i^{mask} = M \odot o_i + (1 - M) \odot f(o_i)$
10:     **end for**
11:     $\theta \leftarrow \theta - \eta \nabla_\theta \mathcal{L}_Q(\theta)$
12:     $\psi \leftarrow (1 - \tau)\psi + \tau\theta$
13: **end for**

---

pixel-level classification layer outputs the probability of each pixel point in two classes (primary and background), where the maximum probability class is the predicted value of that pixel. We transform the pixel-by-pixel class predictions through an interpolation layer into a mask $M \in (0, 1)^{H \times W}$ of the same size as the original image $o \in \mathbb{R}^{H \times W}$. We can use the Hadamard product $\odot : c_{ij} = a_{ij} \times b_{ij}$ to obtain the primary pixels by:

$$o^{mask} = M \odot o + (1 - M) \odot o^{aug}. \tag{6}$$

We use our own constructed *DMC Image Set* for training and testing, more information about image set construction is detailed in Appendix E. The weights of both the encoder and decoder are randomly initialized. After training is completed, the weights of the model are saved as a pre-trained model to be used in the upstream task of RL. We use the cross-entropy loss as the objective function for training the network, and the training process is performed until the loss converges. Detailed hyperparameters for training are given in Appendix C.

### 4.3 DIVERSE DATA AUGMENTATION SELECTION

The mask focusing on the primary pixel position allow us to apply diverse data augmentation methods without changing the image semantics too much. We give an optional set that contains a variety of data augmentation operations. When applying data augmentation to the original observation, one of the data augmentations is randomly selected from this set instead of a fixed one. A variety of different data augmentations acting on background pixel regions can directly and effectively enrich the samples and improve the sample efficiency. This is crucial in RL where sampling is costly. Meanwhile, diverse data augmentation can expand the sample space, allowing the encoder to focus more on consistent subject information between observations and their different augmented observations. We call this method Diverse Data Augmentation (DDA), summarized in Algorithm 1.

### 4.4 DIFFERENTIAL DIVERSE DATA AUGMENTATION SELECTION

We further consider whether we need to keep the primary pixels "intact" (i.e., without using data augmentation). This is because subjects are also perturbed (lighting, color, occlusion, etc.) in real generalized environments. In view of this, we propose to use slight and appropriate data augmentation at the primary pixels and aggressive and diverse data augmentation at the background pixels, i.e., differential data augmentation. Combined with DDA, we propose Differential Diverse Data Augmentation (D3A) to further improve the generalization performance of the agent. We choose random convolution as the default data augmentation method for the original observation. At the same time, based on the semantic-invariant state transformation defined above, we believe that augmented observations satisfying this definition can be forced to guarantee semantic invariance without using masks. This method can further realize differential augmentation on the one hand, and reduce the generation of pictures with too many intermediate processes on the other.

---

**Algorithm 2** Differential Diverse Data Augmentation (D3A)

---

**Input:** Inputs of Algorithm 1 with stabilized training steps $T_s$ and empty initialized deque $\mathbb{L}$ of length $l$

**Output:** Loss $\mathcal{L}_Q$ and updated parameters $\theta, \psi$

  6: Lines 1-6 are consistent with Algorithm 1
  7: **if** $t < T_s$ **then**
  8:     **for** transition $i = 1, 2, \cdots, N$ **do**
  9:         Get the primary position mask $M_i = \phi(o_i)$
10:         $o_i^{mask} = M \odot conv(o_i) + (1 - M) \odot f(o_i)$
11:     **end for**
12: **else**
13:     Calculate $d = \frac{1}{N} \sum_{i=1}^{N} d(o_i^{obs}, o_i^{aug})$ as Equation(3) and add $d$ to $\mathbb{L}$
14:     **for** transition $i = 1, 2, \cdots, N$ **do**
15:         **if** $\mathbb{L}$ is not full **then**
16:             $o_i^{mask} = M \odot conv(o_i) + (1 - M) \odot f(o_i)$
17:         **else**
18:             Obtain the first quartile in the deque $\mathbb{L}$ as the threshold $\varepsilon$
19:             **if** $d < \varepsilon$ **then**
20:                 $o_i^{mask} = f(o_i)$
21:             **else**
22:                 $o_i^{mask} = M \odot conv(o_i) + (1 - M) \odot f(o_i)$
23:             **end if**
24:         **end if**
25:     **end for**
26: **end if**
27: $\theta \leftarrow \theta - \eta \nabla_\theta \mathcal{L}_Q (\theta)$
28: $\psi \leftarrow (1 - \tau)\psi + \tau\theta$

---

In Figure 2, we can find that the Q-value estimation during the early training period is inaccurate and unstable, so it is hard to determine the semantic change by it. We choose to use a queue to save the distance of Q-value within an interval after stabilization and choose the first quartile as the threshold. If the distance between the augmented and original observations computed from a batch of observations at the training step is less than the threshold, the current data augmentation is considered semantically unchanged and can be accepted without mask. Otherwise, we need to obtain the mask to focus on the primary pixels. We call this method Differential Diverse Data Augmentation (D3A), summarized in Algorithm 2. The complete pseudocode is shown in Appendix B. More intuitive effects are detailed in Appendix F.

## 5 EXPERIMENTS RESULTS

We evaluate the sample efficiency, asymptotic performance, and generalization performance of our methods in a set of tasks from the DMControl Generalization Benchmark (DMC-GB), which provides two generalization environments, random color change and adding background video.

**Setup.** We implement our method and baselines using SAC (Haarnoja et al., 2018) as base algorithm. We use the same network architecture and hyperparameters for all methods and adopt the settings of Hansen & Wang (2021). The set of random data augmentations to be selected consists of eight options: random overlay, random conv, random cutout, random cutout color, random grayscale, random color jitter, random pepper, and random blur.

**Baselines.** We benchmark our methods against the state-of-the-art methods including: DrQ, PAD, SODA, SVEA, TLDA. More details on these methods are provided in the Appendix A.We run 5 random seeds, and report the mean and standard deviation of episode rewards.

### 5.1 GENERALIZATION PERFORMANCE

We compare the sample efficiency during training with SVEA to demonstrate the comparable performance of DDA and D3A. Figure 4 shows that DDA and D3A achieve better or equivalent asymptotic performance in the training environment. It is worth proposing that many algorithms are close to

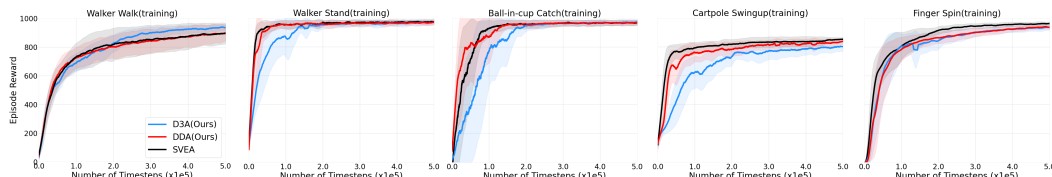

Figure 4: Training Performance. We compare the training performance of DDA, D3A and SVEA on five tasks of DMC, and SVEA chooses random convolution as the data augmentation method.

Table 1: Generalization performance of three settings on the DMC-GB. DDA and D3A outperformed 9 and 12 out of 15 tasks in generalization performance, respectively.

| Setting | DMC-GB | DrQ | PAD | SODA (conv) | SODA (overlay) | SVEA (conv) | SVEA (overlay) | TLDA (conv) | TLDA (overlay) | DDA (Ours) | D3A (Ours) | Δ |
|---|---|---|---|---|---|---|---|---|---|---|---|---|
| color hard | Cartpole Swingup | 682 ±89 | 630 ±63 | 831 ±21 | 805 ±28 | 837 ±23 | 832 ±23 | 748 ±40 | 760 ±60 | 776 ±46 | **831** ±21 | -6(7.2%) |
| | Walker Stand | 770 ±71 | 797 ±46 | 930 ±12 | 893 ±12 | 942 ±26 | 933 ±24 | 919 ±24 | 947 ±26 | 774 ±47 | **968** **±7** | +21(2.2%) |
| | Walker Walk | 520 ±91 | 468 ±47 | 697 ±66 | 692 ±68 | 760 ±145 | 749 ±61 | 753 ±83 | 823 ±58 | 686 +60 | **946** **±8** | +123(15%) |
| | Ball_in_cuo Catch | 365 ±210 | 563 ±50 | 892 ±37 | 949 ±19 | 961 ±7 | 959 ±5 | 932 ±32 | 930 ±40 | 958 ±17 | **970** **±3** | +9(0.9%) |
| | Finger Spin | 776 ±134 | 803 ±72 | 901 ±51 | 793 ±128 | 977 ±5 | 972 ±6 | - | - | 810 ±21 | 970 ±17 | -7(0.7%) |
| | Average | 623 ±119 | 652 ±56 | 850 ±37 | 826 ±51 | 895 ±41 | 889 ±24 | 838 ±45 | 866 ±46 | 801 ±38 | **937** **±11** | +42(4.7%) |
| video easy | Cartpole Swingup | 485 ±105 | 521 ±76 | 474 ±143 | 758 ±62 | 606 ±85 | 782 ±27 | 607 ±74 | 671 ±57 | **848** **±9** | 804 ±34 | +56(7.2%) |
| | Walker Stand | 873 ±83 | 935 ±20 | 903 ±56 | 955 ±13 | 795 ±70 | 961 ±8 | 962 ±15 | 973 ±6 | 971 ±5 | 971 ±3 | -2(0.2%) |
| | Walker Walk | 682 ±89 | 717 ±79 | 635 ±48 | 768 ±38 | 612 ±144 | 819 ±71 | 873 ±34 | 868 ±63 | **927** **±19** | 929 ±15 | +56(6.4%) |
| | Ball_in_cup Catch | 318 ±157 | 436 ±55 | 539 ±111 | 875 ±56 | 659 ±110 | 871 ±106 | 887 ±58 | 855 ±56 | **946** **±47** | 952 ±13 | +65(7.3%) |
| | Finger Spin | 533 ±119 | 691 ±80 | 363 ±185 | 698 ±97 | 764 ±97 | 808 ±33 | - | 744 ±18 | **967** **±10** | 908 ±45 | +159(19.7%) |
| | Average | 578 ±111 | 660 ±62 | 583 ±109 | 811 ±53 | 687 ±101 | 848 ±40 | 832 ±45 | 822 ±40 | **930** **±18** | 912 ±22 | +82(9.7%) |
| video hard | Cartpole Swingup | 138 ±24 | 123 ±24 | - | 429 ±64 | - | 393 ±45 | - | 286 ±47 | **624** **±71** | 454 ±28 | +195(45.5%) |
| | Walker Stand | 289 ±49 | 278 ±72 | - | 771 ±83 | - | 834 ±46 | - | 602 ±51 | **945** **±12** | 894 ±24 | +111(13.3%) |
| | Walker Walk | 104 ±22 | 93 ±29 | - | 381 ±72 | - | 377 ±93 | - | 271 ±55 | **837** **±59** | 564 ±91 | +456(119.7%) |
| | Ball_in_cup Catch | 92 ±23 | 66 ±61 | - | 327 ±100 | - | 403 ±174 | - | 257 ±57 | **856** **±58** | 739 ±95 | +453(112.4%) |
| | Finger Spin | 71 ±45 | 34 ±11 | - | 302 ±41 | - | 335 ±58 | - | 241 ±29 | **813** **+52** | 539 ±78 | +478(142.7%) |
| | Average | 139 ±33 | 119 ±39 | - | 442 ±72 | - | 468 ±83 | - | 331 ±48 | **815** **±51** | 637 ±63 | +347(74.1%) |

Δ = difference between the best of our methods (DDA and D3A) and the best of baselines.
The scores for the best baseline are underlined, while the scores of our methods over the baselines are in bold.

reaching the limits of training performance in some tasks of DMControl, so it is more valuable to focus on the performance of algorithms in various generalization settings.

We evaluate the generalization capability of our method and baselines on three generalization settings of DMC-GB: (1) color hard (randomly change the color of the agent and the background), (2) video easy (the background is replaced with dynamic natural video), (3) video hard (both the floor and the background are replaced with dynamic natural video). Results are shown in Table 1, our methods outperform prior state-of-the-art methods in 12 out of 15 tasks. The results of the baselines are obtained by Hansen & Wang (2021); Hansen et al. (2021b); Yuan et al. (2022a;b) , and "-" indicates that the algorithm has no existing reliable results on this task. In particular, DDA achieves excellent results in the dynamic video generalization setting, especially in the more difficult "video hard" setting with +74.1% improvement on average. This result is mainly based on our use of the mask mechanism that focuses more on the primary position as well as a variety of random augmentations. D3A achieves similar video generalization performance as DDA while making valuable

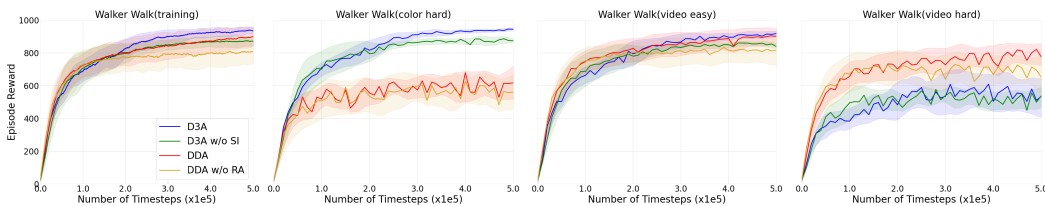

Figure 5: Ablation of the main components (randomized data augmentation and semantic-invariant state transformation) in DDA and D3A. The results show that the green line is lower in performance than the blue line, and the yellow line is lower than the red line.

progress in random color changing environments such as Walker Walk, Walker Stand, and Ball in cup Catch.

## 5.2 Ablation Studies

**Ablation of components.** We perform full ablation experiments on Walker Walk and Finger Spin tasks. Figure 5 shows the results of the training environment for the Walker Walk and the three test environments (color hard, video easy, video hard) for evaluation. Other environments are shown in Appendix D. Among them, DDA (w/o RA) removes the random data augmentation on the basis of DDA, , and only retains the same random convolution as the baseline. The results show that DDA (w/o RA) has a significant degradation in generalization performance, especially in more complex dynamic video environments. The effectiveness of diverse data augmentation for improving the generalization of visual RL is demonstrated. Moreover, in order to validate the significance of differentiated data augmentation, a comparison is made between the performance scores of the D3A (w/o SI) and DDA. The results clearly indicate the necessity of applying different data augmentation to primary and background.

Meanwhile, D3A (w/o SI) removes the original augmented observations that can be left unmasked, i.e., augmented observations that satisfy semantic-invariant state transformation, based on the D3A algorithm. Therefore its performance is reduced compared to D3A. These results support that differential data augmentation can largely improve the generalization capability of the agent.

**Selection of threshold.** The threshold $\varepsilon$, which determines whether a specific data augmentation of the current batch satisfies the semantic-invariant state transformation, is determined by selecting the distance $d$ within the previous window at the current time. We experiment with three choices: the first quartile and meddle value within the window and the use of a threshold of 0. Specifically, the distances computed in the first 40 batches at the current moment are sorted in ascending order, and the first quartile and median values and a threshold of 0 (which is not satisfied by the augmented observations) are chosen, respectively. We finally choose the first quartile as a way of choosing the threshold, and the results of the experiment are presented in Appendix D.

## 6 Conclusion

In this paper, we construct our own DMC Image Set in a manner consistent with human vision by employing $k$-means clustering algorithm to treat images as sets of points. We use this dataset to train a pre-trained segmentation model with an encoder-decoder structure. The purpose of this approach is to enable the agent in RL to focus on primary pixels while avoiding training instabilities caused by inappropriate data augmentation. Meanwhile, the proposed Diverse Data Augmentation (DDA) applies random multiple data augmentations to augment the observation background. This helps the agent encode different augmented views of the same observation consistently. Furthermore, Differential Diverse Data Augmentation (D3A) employs different data augmentations for primary pixels and background pixels. D3A aims to enhance the generalization capability of the agent in complex environments while satisfying the semantic-invariant state transformation that we define. In experiments conducted on the challenging DMControl Generalization Benchmark, our methods demonstrate improved sample efficiency and, more significantly, achieve superior generalization performance compared to the baselines.

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

## A    EXTENDED BACKGROUND

We present details of the extended RL algorithms used for our methods.

**Soft Actor-Critic.** Soft Actor-Critic (SAC) (Haarnoja et al., 2018) is an off-policy actor-critic algorithm that learns a state-action value function $Q_\theta$, a stochastic policy $\pi_\psi$ and a temperature $\alpha$ to find the optimal policy by optimizing a $\gamma$-discounted maximum-entropy objective. The policy evaluation step learns the critic $Q_\theta(o_t, a_t)$ parameters by optimizing the soft Bellman residual of one single step using transitions $\tau_t = (o_t, a_t, o_{t+1}, r_t)$ from an experience buffer $\mathcal{B}$,

$$\mathcal{L}_Q(\theta) = \mathbb{E}_{\tau \sim \mathcal{B}} \left[ (Q_\theta(o_t, a_t) - (r_t + \gamma V(o_{t+1})))^2 \right]. \tag{7}$$

The target value of the next state can be estimated by sampling an action using the current policy:

$$V(o_{t+1}) = \mathbb{E}_{a' \sim \pi} \left[ Q_{\tilde{\theta}}(o_{t+1}, a') - \alpha \log \pi_\psi(a'|o_{t+1}) \right], \tag{8}$$

where $Q_{\tilde{\theta}}$ is a copy of the critic $Q_\theta$ updated using momentum. The policy is learned by minimizing the divergence from the exponential of the soft-Q function at the same states:

$$\mathcal{L}_\pi(\psi) = -\mathbb{E}_{a \sim \pi} \left[ Q_\theta(o_t, a) - \alpha \log \pi_\psi(a|o_t) \right], \tag{9}$$

via the reparameterization trick for the newly sampled action. $\alpha$ is learned against a target entropy.

**Stabilized Q-Value Estimation under Augmentation.** SVEA (Hansen et al., 2021b) is a state-of-the-art off-policy algorithm for visual RL that greatly improves stability of Q-value estimation by only applying augmentation in the current state, without augmenting the next state used for bootstrapping. Instead of learning to predict the target value only from state $o_t$, SVEA minimize a nonnegative linear combination of learning objective over two individual data streams, $o_t$ and $o_t^{aug} = f(o_t, \nu)$ :

$$\mathcal{L}_Q(\theta) = \mathbb{E}_{o_t, a_t, o_{t+1} \sim \mathcal{B}} \left[ \alpha \left\| Q_\theta(o_t, a_t) - y_t \right\|_2^2 + \beta \left\| Q_\theta(o_t^{aug}, a_t) - y_t \right\|_2^2 \right],$$
$$y_t = r_t + \gamma \max_{a_{t+1}} Q_{\tilde{\theta}}(o_{t+1}, a_{t+1}), \tag{10}$$

where $\alpha, \beta$ are constant coefficients that balance the ratio of the unaugmented and augmented data streams (default as $\alpha = \beta = 0.5$ in SVEA).

**Baselines.** We benchmark our methods against the state-of-the-art methods including: (1) **DrQ** (Kostrikov et al., 2021) that applies random shift; (2) **PAD** (Hansen et al., 2021a) that adapts to unseen environments by auxiliary task; (3) **SODA** (Hansen & Wang, 2021) that maximizes the similarity of representations between augmented and original observation; (4) **SVEA** (Hansen et al., 2021b) that applies non-augmented observational calculations Q-target; (5) **TLDA** (Yuan et al., 2022a) that calculates the pixel-by-pixel Lipschitz constant to obtain the relevant pixel positions.

---

Differential Diverse Data Augmentation (D3A)

---

**Input:** random initialized network parameters $\theta, \psi$; candidate data augmentation list $\mathcal{F}$; pre-train segmentation model $\phi$; total number of environment steps $T$; stabilized training steps $T_s$; empty initialized deque $\mathbb{L}$ of length $l$ batch size $N$; target network update rate $\tau$; learning rate $\lambda$; replay buffer $\mathcal{B}$

**Output:** Loss $\mathcal{L}_Q$ and updated parameters $\theta, \psi$

```
 1: for timestep t = 1, 2, · · · , T do
 2:     a_t ∼ π_θ(·|o_t)
 3:     o_{t+1} ∼ P(·|o_t, a_t)
 4:     B ← B ∪ (s_t, a_t, r_t, o_{t+1})
 5:     Sample a batch of size N from B
 6:     Random select augmentation f from F
 7:     if t < T_s then
 8:         for transition i = 1, 2, · · · , N do
 9:             Get the primary position mask M_i = φ(o_i)
10:             o_i^{mask} = M ⊙ conv(o_i) + (1 − M) ⊙ f(o_i)
11:         end for
12:     else
13:         Calculate d = (1/N) Σ_{i=1}^{N} d(o_i^{obs}, o_i^{aug}) as Equation(3) and add d to L
14:         for transition i = 1, 2, · · · , N do
15:             if L is not full then
16:                 o_i^{mask} = M ⊙ conv(o_i) + (1 − M) ⊙ f(o_i)
17:             else
18:                 Obtain the first quartile in the deque L as the threshold ε
19:                 if d < ε then
20:                     o_i^{mask} = f(o_i)
21:                 else
22:                     o_i^{mask} = M ⊙ conv(o_i) + (1 − M) ⊙ f(o_i)
23:                 end if
24:             end if
25:         end for
26:     end if
27:     θ ← θ − η∇_θ L_Q (θ)
28:     ψ ← (1 − τ)ψ + τθ
29: end for
```

---

## B   COMPLETE PSEUDOCODE OF D3A

In Algorithm 2 we provide pseudocode of D3A that omits part of the same process as DDA. To ensure an unambiguous understanding of the algorithm, we provide the complete pseudocode for D3A here.

## C   IMPLEMENTATION DETAILS

We describe in detail the implementation of our algorithm and the hyperparameter setting. For the network architecture of visual RL, we adopt the network architecture of Hansen et al. (2021b) without any modification to the model and hyperparameters to ensure a fair comparison. For the hyperparameter setting of our algorithm, Table 2 shows the detailed hyperparameter values. Table 3 shows the hyperparameter setting for the segmentation models pre-trained using our designed DMC Image Set.

## D   ADDITION EXPERIENCE RESULTS

As stated in Subsection 5.2, random data augmentation and differential data augmentation are proven to be equally effective in Finger Spin, shown in Figure 6.

Figure 7 provides detailed comparison experiments of the threshold selection methods, and it can be found that the first quartile is the most effective, which is also used in D3A.

Table 2: Hyperparameters used in experiments on DMControl.

| Hyperparameter | Value |
| --- | --- |
| Frame rendering | $84 \times 84 \times 3$ |
| Stacked frames | 3 |
| Discount factor | 0.99 |
| Action repeat | 2(finger) 8(cartpole) 4(otherwise) |
| Training step | 500,000 |
| Episode length | 1,000 |
| Batch size | 128 |
| Replay buffer size | 500,000 |
| Optimizer | Adam |
| Actor learning rate | 5e-4(walker walk) 1e-3(otherwise) |
| Critic learning rate | 5e-4(walker walk) 1e-3(otherwise) |

Table 3: Hyperparameters used in pre-train segmentation model.

| Hyperparameter | Value |
| --- | --- |
| Training epoch | 50 |
| Batch size | 4 |
| Optimizer | Adam |
| Learning rate | 5e-5 |
| Decay factor | 0.94 |

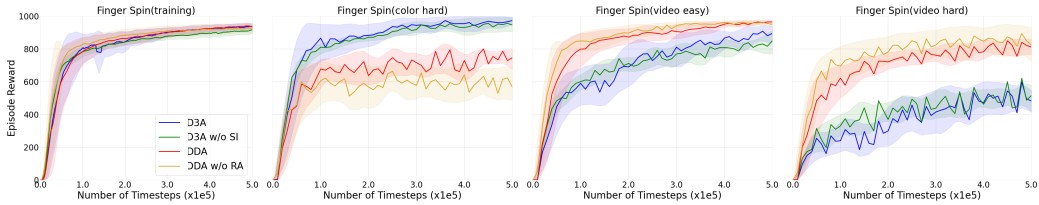

Figure 6: Ablation of the main components (randomized data augmentation and semantic-invariant state transformation) in DDA and D3A. The results show that the green line is lower in performance than the blue line, and the yellow line is lower than the red line.

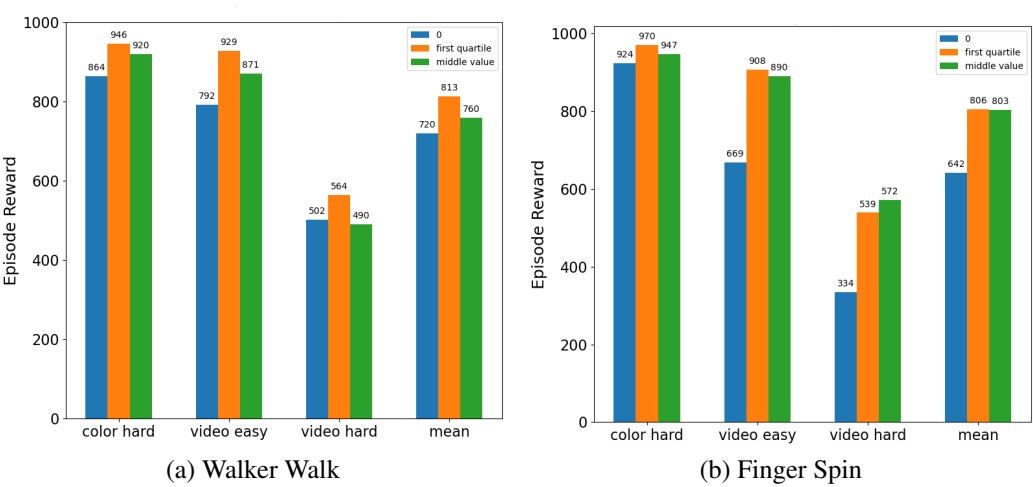

Figure 7: Hyperparameter experiments for threshold selection are performed on Waker Walk and Finger Spin, which can be found to be insensitive. We choose the first quartile as the threshold.

Table 4: Removing the stabilized training step slightly leads to a degradation of the generalization performance

| Setting | DMControl | D3A | D3A (w/o Ts) |
|---------|-----------|-----|--------------|
| color hard | Walker Walk | **946±8** | 923±34 |
| | Finger Spin | **970±17** | 964±16 |
| video easy | Walker Walk | **929±15** | 889±34 |
| | Finger Spin | **908±45** | 863±76 |
| video hard | Walker Walk | **564±91** | 544±49 |
| | Finger Spin | **539±78** | 526±155 |

Table 5: Comparison of generalization performance with more baselines about object-centric and pre-training.

| Setting | DMControl | DDA | D3A | VAI | SGQN | PIE-G |
|---------|-----------|-----|-----|-----|------|-------|
| color hard | Cartpole Swingup | 776±46 | **831±21** | 830±10 | - | 749±46 |
| | Walker Stand | 774±47 | **968±7** | **968±3** | - | 960±15 |
| | Walker Walk | 686+60 | **946±8** | 918±6 | - | 884±20 |
| | Ball_in_cup Catch | 958±17 | **970±3** | 960±8 | - | 964±7 |
| | Finger Spin | 810±21 | **970±17** | 968±6 | - | - |
| video easy | Cartpole Swingup | **848±9** | 802±34 | 761±127 | 761±28 | 587±61 |
| | Walker Stand | **971±5** | **971±3** | 968±2 | 955±9 | 957±12 |
| | Walker Walk | 927±19 | **929±15** | 917±8 | 910±24 | 871±22 |
| | Ball_in_cup Catch | 946±47 | **952±13** | 846±229 | 950±24 | 922±20 |
| | Finger Spin | **967+10** | 908±45 | 953±28 | 956±26 | 837±107 |
| video hard | Cartpole Swingup | **624±71** | 450±28 | - | 569 ± 56 | 401±21 |
| | Walker Stand | **945±12** | 894±24 | - | 851 ± 24 | 852±56 |
| | Walker Walk | **837±59** | 564±91 | - | 739 ± 21 | 600±28 |
| | Ball_in_cup Catch | **856±58** | 739±95 | - | 782 ± 57 | 786±47 |
| | Finger Spin | 813+52 | 539±78 | - | **822 ± 24** | 762±59 |

We define a threshold in D3A for the distance to measure whether an augmented observation satisfies semantic-invariant state transformation. We define a threshold for the distance to measure whether an augmented observation satisfies semantic-invariant state transformation. We conduct evaluation experiments by removing the constraint on the stability training steps (Ts), i.e., Ts = 0, and discuss the role of the stabilized training step. From the results in Table 4, we can see that removing the stabilized training step slightly leads to a degradation of the generalization performance. It is also worth noting that the setting of queue focuses only on the nearby Q-value distance, which may relax the constraints of " recognition of stabilization".

This has some object-centric work Wang et al. (2021); Bertoin et al. (2022)and similar methods that have a pre-training processYuan et al. (2022b). We perform a more extensive performance in Table 5.

# E  DMC Image Set

In computer vision, datasets designed for specific issues are widely available and essential (Deng et al., 2009; Everingham et al., 2015; LeCun et al., 1998). However, to the best of our knowledge there is no public standardized dataset in visual RL for pre-train. The main reasons are: (1) The data required for RL is usually generated by the agent interacting with the environment. (2) The agent trained in the same environment is inclined to overfitting, and the networks of the agent are usually several layers of fully connected networks that do not require more complex networks. However, the tasks where images are used as inputs encounter difficulties in encoding high dimensional data in the first step of representation learning, which gives the opportunity to pre-train for the specific task.

In order to be able to train the model to accurately recognize different regions in an observation, we are motivated by CoCs (Ma et al., 2023) and treat the image as a set of points. Then, a simple clustering algorithm is used for feature extraction based on the color and location information of the observation images. We use 10 tasks selected from DMC-GB. 100 images are randomly generated from the environment, and $k$-means cluster is performed on each image based on pixel features. This is also consistent with human visual perception characteristics. Human vision naturally recognizes the primary part and the background part of an observation based on its features (such as color and position). One of the most important features is the color of the subject. Based on this, for each image, we select the two pixel points with the largest distance in the pixel space as the fixed cluster centers, which is also similar to the fixed cluster center immovable for CoCs. All pixel points of the image are assigned to the clustering centers based on the three-dimensional color features of the pixel points to get the visual subject and background parts, which are represented using 0 and 1, respectively. We call this 0-1 matrix Mask with the same size as the original observation. We filter 770 pairs of images from 1000 pairs of original and clustered images, and randomly select 385 pairs as the train set and 385 pairs as the test set. We call this dataset the DMC Image Set.

We provide raw observations and corresponding Masks for 10 tasks (Cartpole Swingup, Ball in Cup Catch, Finger Spin, Hopper Stand, Pendulum Swingup, Walker Walk, Acrobot Swingup, Cheetah Run, Reacher Easy, Humanoid Run) in the DMC Image Set in Figure 8.

## F    EFFECT OF DDA AND D3A

DMC-GB is a popular benchmark modified from the DMControl Suit introduced by Hansen & Wang (2021) for generalization in visual RL. We conduct validation experiments using five classical tasks, training in a single environment and then evaluating the generalization effect in three different environments such as random color changes and dynamic video backgrounds. In Figure 9, we provide samples of the eight random data augmentations used in our method as well as the effect of DDA and D3A on the original observation.

## G    EXTENSION TO COMPLEX ENVIRONMENTS

We apply the network pre-trained using the DMC Image Set to aDeepMind Manipulation tasks Tunyasuvunakool et al. (2020). We visualize the usefulness of our methods to demonstrate that our methods are theoretically scalable to other tasks in Figure 10.

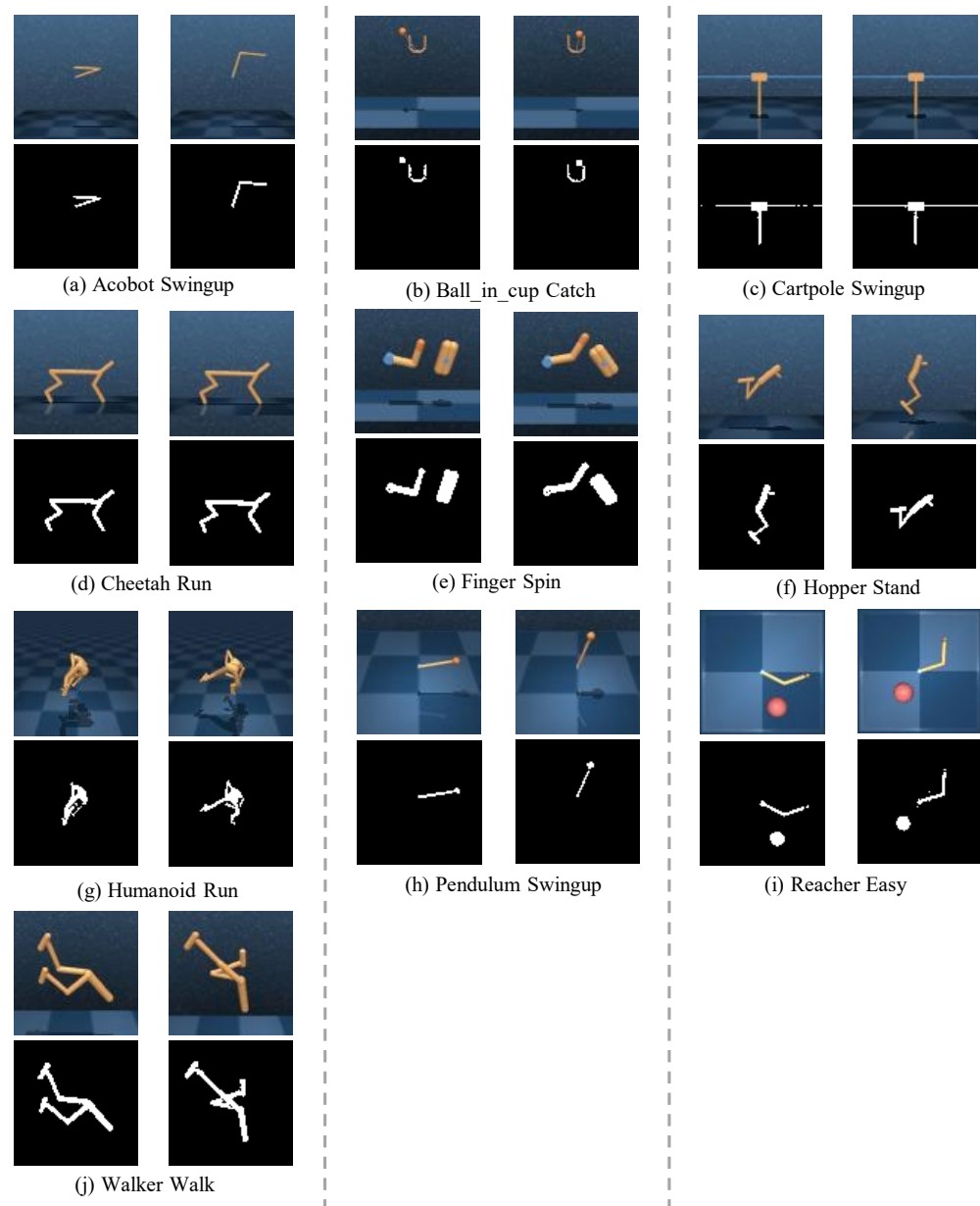

(a) Acobot Swingup
(b) Ball_in_cup Catch
(c) Cartpole Swingup

(d) Cheetah Run
(e) Finger Spin
(f) Hopper Stand

(g) Humanoid Run
(h) Pendulum Swingup
(i) Reacher Easy

(j) Walker Walk

Figure 8: Example image of the DMC Image Set. Each environment shows two pairs of original observations and masks, and the dataset consists of 770 pairs of images from 10 environments.

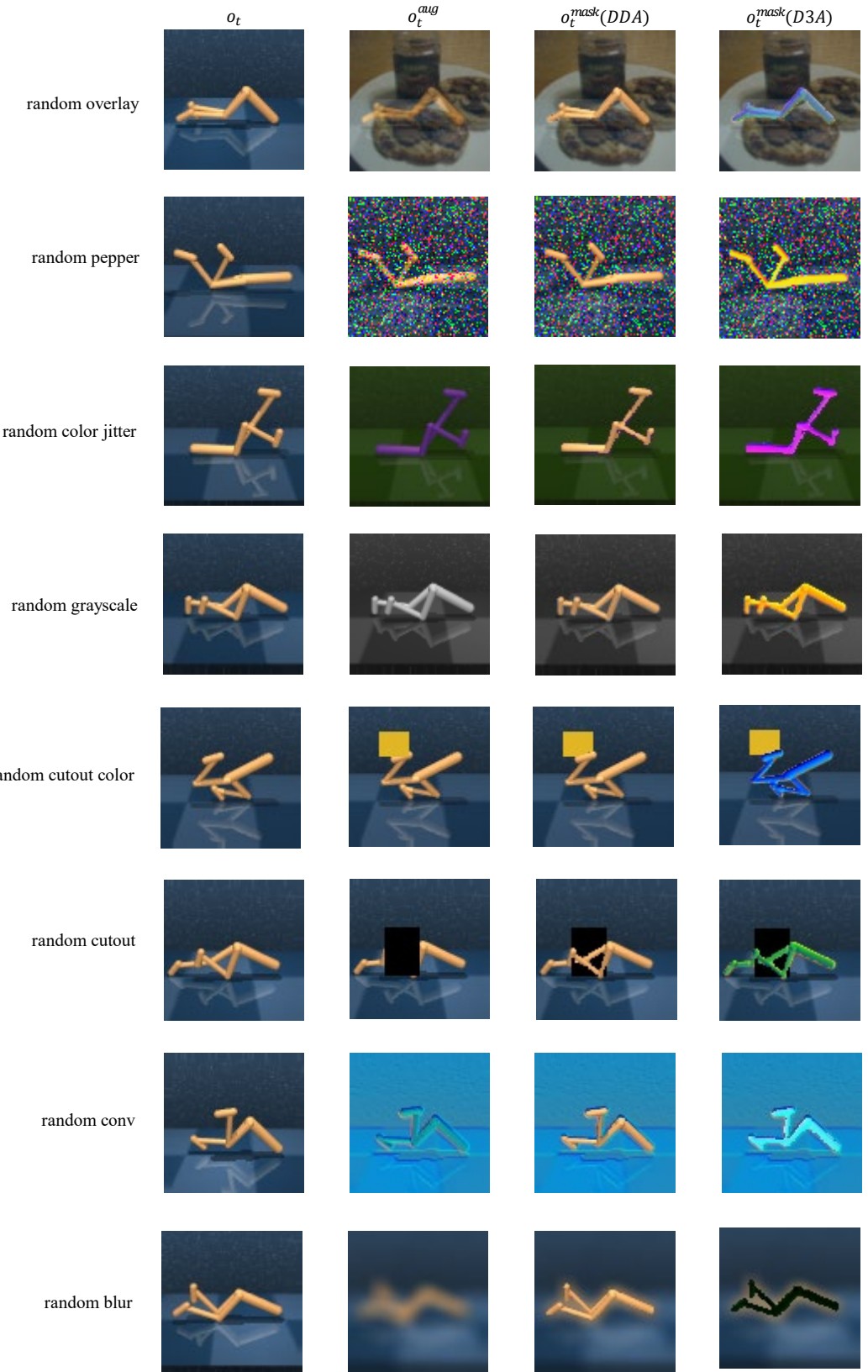

Figure 9: Comparison of naive augmentation with DDA and D3A under 8 augmentation methods.

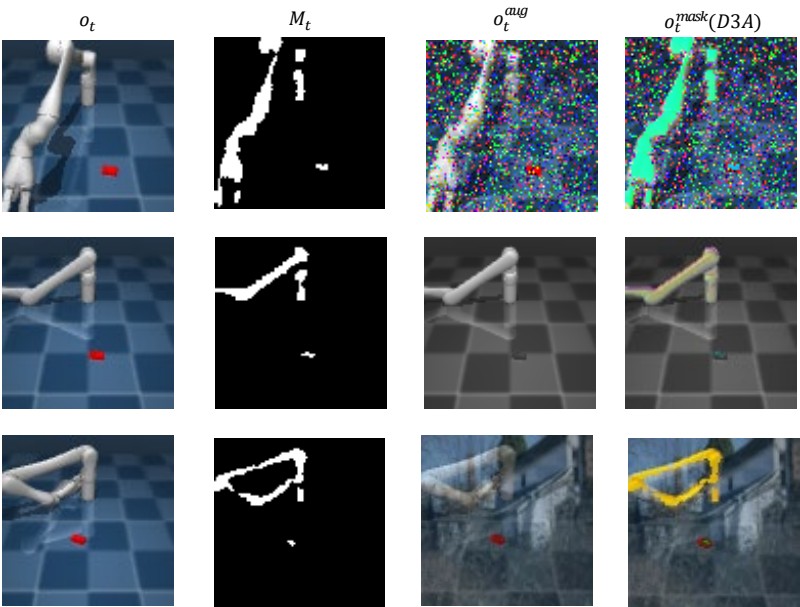

Figure 10: Visual display extended to manipulation task.

