# OpenReview forum: "Focus on Primary: Differential Diverse Data Augmentation for Generalization in Visual Reinforcement Learning"
_ICLR.cc/2024/Conference — Submitted to ICLR 2024_

### Official Review · Reviewer_3m8H · 2023-10-14

**Soundness:** 2 fair
**Presentation:** 2 fair
**Contribution:** 1 poor
**Rating:** 3
**Confidence:** 4

**Summary:**

This work proposes two data augmentation methods DDA and D3A. It utilizes the pre-trained model to get the mask, and uses this mask to produce more appropriate data augmentation for keeping semantic-invariant information.

**Strengths:**

1. D3A can gain better generalization ability than the baselines.

**Weaknesses:**

There are several main problems that I think this paper cannot be accepted:

1.	The pre-trained part is not generalizable. This method overfits to the DMC-GB.  The good performance of D3A relies on the quality of the mask. I think this encoder cannot be applied to any other visual RL tasks.

2.	The main method seems too tricky. The authors do not explain why they choose random conv as a must for augmentation, what about other types of strong augmentation method? Furthermore, the task-specific encoder, the specific augmentation, and some extra introduced hyper-parameters make this paper appear very tricky.

3.    This paper lacks novelty. "Find a proper mask, and keep the primary pixel", I think SGQN [1] is a more general and acceptable method for sloving this problem. I believe that this field should not continue to develop in the direction of proposing better augmentation methods for keeping important pixels.

[1] Bertoin, David, et al. "Look where you look! Saliency-guided Q-networks for generalization in visual Reinforcement Learning."


For writting suggestions:
1. The figures and tables should contain more descriptions, not just a title.

**Questions:**

The questions are mentioned above.

**Details Of Ethics Concerns:**

No ethical concerns.

---

> ### Author Response · Authors · 2023-11-21
> **Official Comment by Authors (Part 1)**
>
> Dear reviewer, thank you for your review and valuable comments on our paper. We have made more detailed explanations and changes to the above issues:
>
> **W1: Generalizability of the pre-training part and dependence of the quality of the mask.**
>
> A1: These issues are key points that we need to further consider and address.
> 1) In our supplementary experiments, the original pre-trained model is applied to the manipulation environment (cf. TLDA [1]). It is quite unfortunate that due to limited time, we have not succeeded in reproducing the baseline. However, we still visualize the usefulness of our methods in the appendix of our paper. This demonstrates that the pre-trained part is not as ungeneralizable as imagined, and is still somewhat transportable in some environments. It is the novel application of the pre-trained segmentation model in visual reinforcement learning in this paper that makes it necessary to manually collect the dataset and pre-train it to demonstrate the feasibility of such an attempt.
> 2) Regarding the issue that the good performance of D3A depends on the quality of the mask, please allow us to emphasize two points. 1) Works with similar ideas, such as TLDA and SGQN, their performance is based on reliable masks. We do not see this as a limitation specific to our approaches. 2) The work in our paper is also not entirely concerned with obtaining a more accurate mask, which is a pursuit of little value. A key to our work lies in the discussion of obtaining a mask faster with less overhead, and in the choice of which data augmentation to employ for the background and key pixels. Most previous works only attempt a weak augmentation to avoid training crashes due to changes in semantics. Our paper explores diverse and differential data augmentation to improve the generalization performance of the agent, which is the key to performance improvement over the baseline.

---

> ### Author Response · Authors · 2023-11-21
> **Official Comment by Authors (Part 2)**
>
> **W2: Complexity of the main method.**
>
> A2: We understand your concerns and viewpoints. Allow us to explain why we chose random convolution as a necessary condition for augmentation. Firstly, random convolution has been widely validated as an effective augmentation method in several works (SODA [2], SVEA [3], TLDA). It is a strong augmentation technique that does not introduce excessive noise, similar to the commonly used weak augmentation (random shift) when initially introducing data augmentation in visual reinforcement learning. Additionally, it is chosen to maintain fairness in comparing training performance with the baseline SVEA, as random convolution is its default data augmentation method. Furthermore, in designing our method, we considered the use of slight and semantically unaltered augmentations in the primary pixels, and random convolution is a suitable choice that fulfills this requirement.
>
> Regarding the complexity of the main method, we would like to discuss it from three points:
> 1) We are very worried about that Figure 3 showing both DDA and D3A, which are not identical algorithms, has given you the impression that the methods are more complex. In reality, neither DDA nor D3A is complex. You can simply divide the algorithms into two processes, generating masks and data augmentation selection. In the first process, the actual segmentation network is more lightweight compared to semantic segmentation networks commonly used in computer vision, and the pre-training process can also be done in a few minutes. Considering the high cost and overhead of similar works through complex computations, the methods in our paper greatly improve the training speed and performance.
> 2) Data augmentation selection: our main point of view is that we would like to introduce more diverse data augmentations and differentially operate on the background and primary pixels under the stabilization of learning. The set containing 8 data augmentation methods and the default random convolution used in the methods are an attempt based on previous works, not purposely customized.
> 3) The only hyperparameters in the method are the choice of the threshold in the Q-value queue and the stabilized training step. For the selection of the threshold, we have already demonstrated its insensitivity in the ablation experiments of the paper. The discussion of the stabilizing training step is also raised by reviewer XRib. However, as shown in Fig. 2, the Q-value is inaccurate in the pre-training phase, and the identification of " stabilization " is difficult due to the different training processes. We chose to implement D3A at a late time (200k on the total 500k steps), which makes the effect of D3A shorter but still significant. It is also worth noting that the setup of the queue L makes it possible to focus only on the proximity of the Q-value distance d, which may relax the restriction of "stabilization", and we remove the restriction of the stabilization training step (Ts), i.e., Ts = 0, for the evaluation experiments. From the results in Table 4, we can see that removing the stabilized training step slightly leads to a degradation of the generalization performance. More details of the training we have added to the appendix of our paper.
>
> **W3: Lack of novelty.**
>
> A3: We greatly value your perspectives on the development of this field.
> 1) Although our methods may share some similarities with SGQN in some aspects, the goal of our research is to explore a new data augmentation option to provide better generalization performance in reinforcement learning tasks. In visual reinforcement learning, mask-based work (VAI [4], SGQN, TLDA, and even OCR methods, as mentioned in the discussions of the other three reviewers, respectively) have all been proposed in the last two years, and we believe that there is still room for improvement and new innovations.
> 2) In particular, our methods not only focus on how to efficiently obtain accurate masks but also focus on repurposing many data augmentation methods that are considered inapplicable and introduce the idea of differentially augmenting primary pixels from background pixels after masking.
> 3) The proposed semantic-invariant state transformation is also a further addition to optimality-invariant state transformation, a definition that all data augmentation-based reinforcement learning ideally follows. We hope that these works contribute to the development of the field.

---

> ### Author Response · Authors · 2023-11-21
> **Official Comment by Authors (Part 3)**
>
> **W4: More descriptions of figures and tables.**
>
> A4: We have further modified the figures and tables based on your suggestions by providing more detailed descriptions below the title to explain what is presented. Please also refer to our latest PDF submission for specifics.
>
> We sincerely hope that our response has addressed your questions and will contribute to improving the final score of the paper. If you have any further questions or suggestions, please don't hesitate to let us know. We would be more than happy to provide clarification and answer any additional inquiries you may have.
>
> [1] Don’t Touch What Matters: Task-Aware Lipschitz Data Augmentation for Visual Reinforcement Learning. IJCAI 2022.
>
> [2] Generalization in Reinforcement Learning by Soft Data Augmentation. ICRA 2021.
>
> [3] Stabilizing Deep Q-Learning with ConvNets and Vision Transformers under Data Augmentation. NeurIPS 2021.
>
> [4] Unsupervised Visual Attention and Invariance for Reinforcement Learning. CVPR 2021.

---

> ### Comment · Reviewer_3m8H · 2023-11-22
> **Official Comment by Reviewer 3m8H**
>
> Thanks for the authors' reply. I acknowledge the efforts made by the authors.  I list my main concerns here:
>
> 1. In addition to the manipulation tasks in DMC, there are other robotics, manipulation, or locomotion environments worth exploring. Actually, I would like to see the results on different benchmarks beyond DMC. The reason is that working exclusively on DMC environment might lead to the encoder overfitting to the DMC background.
>
> 2. It's hard to believe that an encoder pre-trained only on DMC would perform well on other tasks and environments. It doesn't make sense.
>
> 3. While D3A runs more efficiently, its speed is primarily due to the use of a pre-trained DMC encoder. Besides, there are many segmentation models in vision. I think you should compare with these models to show the effectiveness of your method. Otherwise, you cannot convince the readers.
>
> 4. Numerous methods exist for obtaining masks, but I believe using domain-specific pre-training to derive masks lacks elegance and novelty. If you want to enhance the augmentation, you should focus on other methodology, i.e., . SRM [1]. Otherwise, it's hard to reach the acceptance standard of ICLR.
>
> 5. The concept of semantic-invariant state transformation is a simple variation of optimality-invariant state transformation. This idea is well-known in the field, and this paper merely formalizes it, which does not constitute novelty.
>
> [1] Spectrum Random Masking for Generalization in Image-based Reinforcement Learning.  NeurIPS 2022
>
> Considering the above concerns, I would maintain my score.

---

### Official Review · Reviewer_xszw · 2023-10-30

**Soundness:** 3 good
**Presentation:** 3 good
**Contribution:** 2 fair
**Rating:** 5
**Confidence:** 3

**Summary:**

This paper proposes two novel approaches, Diverse Data Augmentation (DDA) and Differential Diverse Data Augmentation (D3A), to address the challenges of generalization and sample efficiency in visual reinforcement learning. The methods leverage a pre-trained encoder-decoder model to segment primary pixels and avoid inappropriate data augmentation. DDA focuses on the consistency of encoding to improve generalization, while D3A further enhances generalization by using proper data augmentation for primary pixels while maintaining semantic-invariant state transformation. Extensive evaluation on DeepMind Control Suite tasks demonstrates significant improvements in the agent's generalization performance in unseen environments and increased sample efficiency of off-policy algorithms.

**Strengths:**

The introduction of Differential Diverse Data Augmentation is quite intriguing, and the method itself is intuitive and straightforward.

**Weaknesses:**

1. The author mentioned the use of a clustering algorithm for image segmentation but did not clarify how these images for clustering were collected. Was a random strategy employed for data collection?

2. There are many object-centric works [1,2,3] that are quite similar to this paper. It would be good if the authors could highlight the difference. I am also curious to know if the proposed method would have an advantage over other object-centric methods.

  [1]Unsupervised Visual Attention and Invariance for Reinforcement Learning. CVPR 2021.

  [2] Look where you look! Saliency-guided Q-networks for generalization in visual Reinforcement Learning. NeurIPS 2022.

  [3] An Investigation into Pre-Training Object-Centric Representations for Reinforcement Learning. ICML 2023.

3. The experiments lack ablation studies on certain hyperparameters. For example, what criteria were used to determine the stabilized training steps? After all, different environments and tasks would require different training conditions.

4. There is a lack of comparison with other pretraining methods. After all, this study utilizes pretraining, while the comparison methods are all end-to-end approaches, which may not be entirely fair.

**Questions:**

Please refer to the weaknesses.

---

> ### Author Response · Authors · 2023-11-21
>
> Dear reviewer, thank you for your review and valuable comments on our paper. We have made more detailed explanations and changes to the above issues:
>
> **W1: How image data is collected.**
>
> A1: Thank you for your attention to how the images used for clustering are collected. In fact, during the data collection process, we obtain different observed images directly by resetting the environment. This is the same essence as using a random policy, i.e., it is independent of the learning process of reinforcement learning.
>
> **W2: Differences and advantages with other works.**
>
> A2: You mentioned some object-centric works (VAI, SGQN, OCR) that are similar to our research, and we offer the following discussion of the differences and strengths of our methods with respect to the core of these works:
> 1) VAI uses unsupervised learning to filter task-irrelevant background information and retain foreground information. The starting point is similar to our methods.  VAI needs to train Encoder, Decoder, and KeyNet to acquire feature maps, reconstruct images, and capture foreground key points, respectively. However, VAI is based on the assumption that the changes in the foreground are much larger than the changes in the background, which may be limiting in many cases.
> 2) SGQN is computed by guided backpropagation to obtain a binary attribution map (mask) of the original observation. Using this map as a target, a decoder is added to converge to that target, yielding a decoder that predicts the augmented observation attribution map.
>
> Our methods are in line with the previous two works regarding the use of masks to shield background interference to maintain training stability. However, two differences are that the way of obtaining the mask is different and that they do not have our discussion on differential and diverse data augmentation. Our advantages also include two points: our methods avoid complex computations and time efficiency is efficient. Also, the performance is outstanding under the same test benchmarks, for a detailed comparison we provide the data in Table 2.
>
> 3) The third is an experimental work. It investigates the effectiveness of Object-Centric Representation (OCR) pre-training as a framework for RL representation learning through empirical experiments, and presents a benchmark evaluating several current OCR pre-training algorithms.
>
> Our methods also belong to a simple OCR in nature. Although the experimental environment of this paper focuses more on visually simple 2D geometric multi-object scenes, we also strongly endorse the conclusions obtained in this paper: the effectiveness of OCR pre-training for RL generalization and inference tasks, the continued improvement of OCR models studied in more complex and realistic environments.
>
> **W3: How to determine the stabilized training step.**
>
> A3: A similar problem is pointed out by reviewer XRib (W2&A2). Briefly, what we do is to visualize the change of Q-value during training for several sets of environments, choosing an approximate stabilization time. Maybe this is imperfect but simple and intuitive.
>
> **W4: Comparison with other pre-training methods.**
>
> A4:
> 1) Our pre-trained model is only used to generate masks which is not directly involved in the subsequent reinforcement learning. So it is independent of the encoder of the policy and Q-function. Therefore, compared to other methods of calculating masks, the impact on the performance of generating masks through the network is small, but the reduction in training cost is huge.
> 2) The improvement in the generalization performance of our methods is due to differential and diverse data augmentation. This part is equally transferable and effective to other mask-based methods. To fully allay your concerns about a fair comparison, we added PIE-G [1] as a baseline in Table 2, which is a method that uses ResNet as a pre-trained image encoder. The comparison shows that our method still takes the lead in generalization performance.
>
> We sincerely hope that our response has addressed your questions and will contribute to improving the final score of the paper. If you have any further questions or suggestions, please don't hesitate to let us know. We would be more than happy to provide clarification and answer any additional inquiries you may have.
>
> [1] Pre-Trained Image Encoder for Generalizable Visual Reinforcement Learning. NeurIPS 2022.

---

### Official Review · Reviewer_ziWN · 2023-10-31

**Soundness:** 3 good
**Presentation:** 3 good
**Contribution:** 3 good
**Rating:** 5
**Confidence:** 3

**Summary:**

This paper proposes a method to better augment visual inputs for RL. The method relies on a simple model that produces a mask that selects the main object in the image (in this case the agent), and then augments the background and the main object differently. The proposed DDA doesn't augment the main object, while D3A introduces an adaptive strategy that, depending on how much the outputs of the model change, decides whether to augment the main object or not.
The method is tested on Deep Mind Control suite, and achieves impressive performance in the setting with added perturbations.

**Strengths:**

- The paper is quite clearly written
- The proposed achieves strong performance against the baselines on DMC tasks.
- The idea of using quartiles for epsilon avoids having an additional hyperparameter for D3A.

**Weaknesses:**

- The method is only tested on DMC. While it achieves impressive performance, more tasks would be needed to see if the idea has merit.
- The method is quite complicated, requiring an additional module to produce the mask.
- The method is quite close to TLDA [1], and is not tested as extensively as TLDA

comments:
- Minor suggestion: can you highlight the best baseline in Table 1, like underscore it for clarity?
- Section 4.4 "without being used a mask" sounds weird
- Algorithm 2, lines 26, 27: should this be outside the big if? I understand Algorithm 2 refers to Algorithm 1 to save space. It would be useful to have a full version in the appendix to avoid confusion.

I'm willing to raise my score if authors provide additional evaluation in another environment (e.g. robotic manipulation).

[1] TLDA: Don't Touch What Matters: Task-Aware Lipschitz Data Augmentation for Visual Reinforcement Learning https://arxiv.org/abs/2202.09982

**Questions:**

1. How would the method handle more complicated, real-world environments where obtaining a mask is not as easy? In DMC, clustering the main object is fairly easy, while in more cluttered scenes it will be more complicated. This taps into a whole new area of research on segmentation, but I want to know if authors have thought about this.

---

> ### Author Response · Authors · 2023-11-21
> **Official Comment by Authors (Part 1)**
>
> Dear reviewer, thank you for your review and valuable comments on our paper. We have made more detailed explanations and changes to the above issues:
>
> **W1: More training and evaluation on another environment (robotic manipulation).**
>
> A1: Thank you very much for acknowledging the impressive performance of our method on DMC, and we appreciate your positive feedback. We have also conducted additional experiments to further evaluate our method. We have applied our methods to the DMC manipulation task (cf. TLDA). It is quite unfortunate that due to limited time, we have not succeeded in reproducing the baseline. However, we still visualize the usefulness of our methods in the appendix of our paper, demonstrating that our methods are theoretically scalable to other tasks.
>
> **W2: Additional Module.**
>
> A2:
> 1) We are very worried about that Figure 3 showing both DDA and D3A, which are not identical algorithms, has given you the impression that the methods are complex. In reality, neither DDA nor D3A is complex. You can simply divide the algorithms into two processes, generating masks and data augmentation selection. In the first process, the actual segmentation network is more lightweight compared to semantic segmentation networks commonly used in computer vision, and the pre-training process can also be done in a few minutes. Considering the high cost and overhead of similar works through complex computations, the methods in our paper greatly improve the training speed and performance.
> 2) Additionally, the RL framework tends to have a lightweight network architecture. Requiring a lightweight network to perform complex generalization tasks is not a reasonable approach. Methods for solving this kind of visual generalization problem (TLDA, SGQN [1], CCLF [2], etc.) inevitably introduce new auxiliary tasks or computational modules. The modules added in our methods are extremely cost-effective in terms of size and training time compared to other methods, and we hope you do not see this as a weakness of ours.
>
> **W3: Differences with TLDA.**
>
> A3:
> 1) TLDA serves as an important inspiration and baseline for our work in this paper. However, it has significant limitations in terms of unacceptable computational complexity, training duration, and large GPU memory requirements. We have made significant improvements in addressing these issues in the mask-generation process. Additionally, our methods incorporate differential and diverse data augmentation techniques for background and primary pixels that are not available in TLDA, which is the key to the performance improvement of our methods.
> 2) It should be emphasized that almost all reviewers paid extra attention to the pre-training process and the mask-generation process of the segmentation network. In fact, these processes are important foundations of the methodology of our paper, but they are not the key to be discussed at the core. A more important point of our paper lies in the impact of differential and diverse data augmentations on the generalization performance after separating primary and background pixels using masks.
>
> **C1: Modifications to Table 1.**
>
> A1: Regarding highlighting the optimal baseline in Table 1, we fully agree with you and believe that this will help the reader understand the results more clearly. Having accepted your suggestion, we have modified Table 1 to highlight the results of the optimal baseline using underlining. The necessary notes have also been added to the endnotes of the table.
>
> **C2: Expression issues in section 4.4.**
>
> A2: The expression problem you pointed out does lead to unclear sentences. We apologize for this and have revised Section 4.4--” we believe that augmented observations satisfying this definition can be forced to guarantee semantic invariance without using masks.”
>
> **C3: The complete pseudo-code for Algorithm 2.**
>
> A3: The issue you mentioned regarding the placement of lines 26 and 27 in Algorithm 2 is indeed an issue that needs to be clarified. To address this issue and to avoid confusion, we provide the full version of Algorithm 2 in the Appendix so that the reader can better understand and use the algorithm.

---

> ### Author Response · Authors · 2023-11-21
> **Official Comment by Authors (Part 2)**
>
> **Q1: How to deal with more complex real-world environments.**
>
> A1: In our study, we recognize that for more complex real-world environments, there may be some challenges in obtaining masks. However, what we would like to discuss in more depth is the feasibility of the approach based on segmentation networks, which avoids complex computations and high training costs. Also, a focus of the paper's discussion is the impact on the generalizability of different data augmentation choices taken for different pixels after acquiring the mask. At the same time, our methods can be combined with advanced segmentation techniques to address these challenges. We believe that combining reinforcement learning with advances in the field of segmentation research can lead to better results in more complex real-world environments.
>
> We sincerely hope that our response has addressed your questions and will contribute to improving the final score of the paper. If you have any further questions or suggestions, please don't hesitate to let us know. We would be more than happy to provide clarification and answer any additional inquiries you may have.
>
> [1] Look where you look! Saliency-guided Q-networks for generalization in visual Reinforcement Learning. NeurIPS 2022.
>
> [2] CCLF: A Contrastive-Curiosity-Driven Learning Framework for Sample-Efficient Reinforcement Learning. IJCAI 2022.

---

> > ### Comment · Reviewer_ziWN · 2023-11-22
> > **Response**
> >
> > Thank you for addressing my comments! My concern about complexity has been addressed. Still, unfortunately, I choose to keep my score unchanged. I do see promise in this method, but I think having one or two more tasks where the method demonstrates good performance is necessary. Since some components of the system require environment-specific data and masking module, it's important to show that the idea has merit in other settings.

---

### Official Review · Reviewer_XRib · 2023-10-31

**Soundness:** 2 fair
**Presentation:** 1 poor
**Contribution:** 2 fair
**Rating:** 3
**Confidence:** 5

**Summary:**

The paper introduces a novel technique named Diverse Data Augmentation (DDA) and its enhanced variant, Differential Diverse Data Augmentation (D3A), tailored for targeted data augmentation in image-based Reinforcement Learning.
Their approach is based on is the utilization of a segmentation network (Segnet) which is trained on a custom dataset to discern between foreground and background pixels in the observations. The authors leverage the predictions from this segmentation network to generate masks, facilitating the application of strong data augmentation to the background pixels. This ensures that the augmented observation undergoes only minimal semantic alterations. In the D3A variant, the authors incorporate a milder form of data augmentation on the foreground pixels, but only after confirming that the Q-value estimations between the augmented and original observations aren't drastically different. Both methods are applied using the SVEA framework: the augmented observations are exclusively presented to the Critics, leaving the target Critics unexposed to them. The efficacy of both methods is confirmed through empirical experiments on the DeepMind Distracting Control suite benchmark.

**Strengths:**

*    The authors have devised an original strategy of applying differential data augmentation: intense augmentation on non-critical pixels and milder augmentation on task-relevant pixels. This layered approach offers the potential for enhancing robustness without overwhelming the primary information in the images.

*    The paper introduces a criterion to determine if an augmented observation should be incorporated during the training process. Such a selective approach aims that only beneficial augmented data contributes to the learning, potentially reducing noise and improving convergence.

*    The method's results on the DeepMind Distracting Control suite benchmark provide evidence of its practical utility. While limited to this benchmark, it's a step towards validating the approach's applicability in certain environments.

**Weaknesses:**

**Weaknesses**

- A notable dependency of the method is its reliance on the Segnet network, specifically trained on a custom dataset crafted by the authors. This dataset facilitates supervised learning to distinguish between background and task-relevant pixels. The intensive human supervision required to curate this dataset raises concerns about the method's scalability and adaptability to more intricate environments.

- The approach necessitates the identification of a threshold, determining when the Q-values of augmented observations deviate substantially from the original observations' Q-values. The definition of this threshold hinges on some form of "stabilization" during training. The paper would benefit from a more thorough discussion regarding the identification and practical implications of this "stabilization."

- The experimental results suggest that the methods might be overly tailored for the specific benchmark in question. For instance, DDA demonstrates superior performance on distracting backgrounds due to its emphasis on robust background augmentation, while D3A outperforms on color distractions that modify task-relevant object colors. Such specificity could limit the method's generalization across diverse settings.

- The problem formulation section appears convoluted and would benefit from a more coherent presentation.

- The ablation study lacks clarity in specifying the particular distracting setting upon which the performance metrics are based. Given the distinct performances of DDA and D3A under varying distracting scenarios, this omission is significant. Additionally, the paper doesn't provide clarity on what constitutes "DDA without Random Augmentation." Is it merely SAC, or DDA with a predetermined data augmentation type? If it's the latter, the specific augmentation type ought to be explicitly mentioned.

- While the authors claim to apply their method to SAC, in reality, the application is more in line with the SVEA framework, as there's a shared emphasis on excluding data augmentation from target critic estimations.

- Several parts of the paper are marred by ambiguous language, unclear expressions, and typographical errors. Examples of such problematic statements include: "expanding the latent sample space," "migrate the trained representations to tasks for visual driving," and "we define an transformation.", "+339% improvement" .

The paper would undoubtedly benefit from a thorough editorial review to rectify these inconsistencies and improve overall clarity.

**Questions:**

Regarding the DDA approach, when a particular masked augmented observation is rejected based on Q-values estimation, why opt for using the original augmented observation instead of a masked one like in DDA? What motivated this design choice?

---

> ### Author Response · Authors · 2023-11-21
> **Official Comment by Authors (Part 1)**
>
> Dear reviewer, thank you for your review and valuable comments on our paper. We have made more detailed explanations and changes to the above issues:
>
> **W1: Dependence on a custom dataset and intensive manual supervision.**
>
> A1: We fully understand your concerns regarding the dependency and scalability of our method. Indeed, the manual supervision required to manage this dataset is not intensive and is also scalable.
> 1) The Segnet-based network is chosen in our research because it demonstrates excellent performance in our task, with a simple network structure and effective discrimination between background and task-related pixels. Our dataset consists of a small number of randomly collected images from the environment. Unlike traditional methods, we employ intuitive color clustering for labeling, without introducing excessive manual supervision. Although this process has limitations, it yields significant results in terms of time and effectiveness. Compared to the high computational cost and time overhead of similar methods, we attempt to leverage the segmentation network which is a pioneering work, and more accurate segmentation algorithms would support the applicability of our methods in more complex environments.
> 2) We apply the network pre-trained using this dataset to a more complex environment (robotic manipulation). It is quite unfortunate that due to limited time, we have not succeeded in reproducing the baseline. However, we still visualize the usefulness of our methods in the appendix of our paper, demonstrating that our methods are theoretically scalable to other tasks.
>
> **W2: Discussion on "Stabilization".**
>
> A2: The concept of " stabilization " is abstract, and in the paper, we propose a "Semantic-Invariant State Transformation" as a rephrasing of "Optimality-Invariant State Transformation". The underlying idea is not to require the Q-values of the augmented and original observations to be exactly the same, as this is impossible to achieve.
> 1) We define a threshold for the distance to measure whether an augmented observation satisfies semantic-invariant state transformation. However, as shown in Figure 2, Q-values are inaccurate in the early stages of training, making it challenging to identify " stabilization " based on different training processes. We chose a relatively late time step (200k on the total 500k steps) to implement D3A, which reduces the duration of D3A's effect but still produces significant results.
> 2) It is also worth noting that the setting of the queue focuses only on the nearby Q-value distance, which may relax the constraints of " recognition of stabilization ". We conduct evaluation experiments by removing the constraint on the stability training steps (Ts), i.e., Ts = 0, and discuss the role of the stabilized training step. From the results in Table 4, we can see that removing the stabilized training step slightly leads to a degradation of the generalization performance. More details of the training we have added to the appendix of our paper.
>
> **W3: Over-tailoring for specific benchmarks.**
>
> A3: Firstly, similar practices have been employed in SODA, SVEA, and TLDA, where two different data augmentation methods are used to improve the scores in generalization. Specifically, random conv aids in color changes, while random overlay aids in video backgrounds. This point is explicitly stated in the baseline. Thus, DDA and D3A, also as algorithmic variants, are not unfairly compared. On the contrary, individual algorithms can also achieve good results in different environments. For example, DDA outperforms the baselines in 9 out of 15 environments, and D3A outperforms the baselines in 12 out of 15 environments.
>
> **W4: Simplification and coherence of the problem formulation section.**
>
> A4: We sincerely appreciate your feedback, and we will carefully review the problematic sections of the paper and take measures to ensure they are more coherent and understandable. For more details on the modifications, please check the latest version of the PDF.
>
> **W5: Specific distracting setting and the specific meaning of DDA (w/o RA).**
>
> A5: We conducted uniform experiments in the ablation study across three generalization settings (color hard, video easy, video hard), and please refer to Section 5.2 and Figures 5-7 for the results. The consistency of the experimental results demonstrates the roles of the components in DDA and D3A. If you have any further questions, we would be more than happy to address them.
>
> We sincerely acknowledge your feedback regarding the insufficient explanation of "DDA without Random Augmentation" in our paper. "DDA without Random Augmentation" refers to the absence of random data augmentation while retaining the default random convolution used in several baselines. DDA (w/o RA) can be further understood as SVEA (random conv) + segmentation network for the mask. We explicitly mention this point in the latest paper to avoid any ambiguity.

---

> ### Author Response · Authors · 2023-11-21
> **Official Comment by Authors (Part 2)**
>
> **W6: The framework of methods.**
>
> A6: Thank you very much for your careful review. Indeed, the approach presented in this paper is based on the SVEA framework with modifications, and the experimental section thoroughly compares the results with SVEA. Additionally, the listed baselines all utilize SAC as the framework.
>
> **W7: Details of writing.**
>
> A7: We acknowledge that there may be some ambiguous language and unclear expressions in the paper. We have carefully reviewed each section of the paper to identify any inaccuracies and typographical errors and make the necessary corrections to ensure more precise, clear, and unambiguous language. Regarding the examples you mentioned, we have highlighted the following modifications:
> 1) Data augmentation can generate more equivalent data (samples) without substantially increasing the amount of data. In representation learning, this is commonly referred to as "high-dimensional sample space" and "low-dimensional latent space/feature space." We indeed do not strictly differentiate between the use of "latent" and "unseen" in our paper. We correct it to "expanding the sample space."
> 2) “migrate the trained representations to tasks for visual driving.” has been changed to: "transfer the pre-trained representations to the reinforcement learning task of visual driving.
> 3) The typographical errors have been corrected: "we define a transformation" and "+74.1% improvement."
>
> **Q1: Justification for data augmentation choices in different situations.**
>
> A1: Thank you very much for your attention to this detail, I understand that you want to ask what contributes to the fact that we choose to use the original augmented observation rather than the masked observation in D3A when the distance based on the Q-value estimation is less than the threshold (i.e., when the augmented observation satisfies the semantic-invariant state transformation). The starting point of this approach is that we believe that augmented images that satisfy our definition of semantic invariant state transformation will not lead to training instability and changes in the semantic information. In this case, any previous mask is redundant. Instead, we believe that this approach reduces the number of computations of masks and guarantees the improvement in generalization performance that comes with data augmentation. We appreciate the opportunity to clarify this aspect of our approach. If you have any further questions, please feel free to ask.
>
> We sincerely hope that our response has addressed your questions and will contribute to improving the final score of the paper. If you have any further questions or suggestions, please don't hesitate to let us know. We would be more than happy to provide clarification and answer any additional inquiries you may have.

---

> ### Comment · Reviewer_XRib · 2023-11-22
>
> I acknowledge the author's response to my question. However, after considering this information and the reviews provided by my colleagues, I am maintaining my previous evaluation and will not change my score.

---

### Meta-Review · Area_Chair_XgcW · 2023-12-03

**Metareview:**

The paper proposes an improved data augmentation method for training RL agents.
* The paper claims that not all data augmentations are helpful, as some may actually reduce useful information about the image (e.g. accidentally randomly blocking out key parts of an image).
* The authors then use an image segmentation model to provide an extra mask to the Q-function during training to help it identify the important features.
* Experiments were conducted over DM-Control and demonstrates that the proposed method nearly unanimously improves rewards.

While the paper demonstrates that this method is potentially viable, all reviewers raised concerns about:
* Generality of method. Currently, experiments are only conducted on DM-Control, and not other environments where it may be harder to segment "important features".
* Method's complication: It requires pretraining over user-created labelled data and leads to additional hyperparameters.

As-is, the paper is not sufficient for acceptance. I would recommend the authors to address these key concerns and resubmit for another venue.

**Justification For Why Not Higher Score:**

The paper does not raise any "red flags", but the reviewers unanimously agreed that the method might be too DM-Control specific and complex.

**Justification For Why Not Lower Score:**

N/A

---

### Decision · Program_Chairs · 2024-01-16

Reject